# Intranasal HBsAg/HBcAg-Containing Vaccine Induces Neutralizing Anti-HBs Production in Hepatitis B Vaccine Non-Responders

**DOI:** 10.3390/vaccines11091479

**Published:** 2023-09-12

**Authors:** Kana Shiraishi, Osamu Yoshida, Yusuke Imai, Sheikh Mohammad Fazle Akbar, Takahiro Sanada, Michinori Kohara, Takashi Miyazaki, Taizou Kamishita, Teruki Miyake, Masashi Hirooka, Yoshio Tokumoto, Masanori Abe, Julio Cesar Aguilar Rubido, Gerardo Guillen Nieto, Yoichi Hiasa

**Affiliations:** 1Department of Gastroenterology and Metabology, Graduate School of Medicine, Ehime University, Toon 791-0295, Japan; kanasbaishi@gmail.com (K.S.); yskyskimai@yahoo.co.jp (Y.I.); sheikhmohammadfazle@gmail.com (S.M.F.A.); miyake.teruki.mg@ehime-u.ac.jp (T.M.); masashihirooka@gmail.com (M.H.); yotoku@m.ehime-u.ac.jp (Y.T.); masaben@m.ehime-u.ac.jp (M.A.); hiasa@m.ehime-u.ac.jp (Y.H.); 2Department of Microbiology and Cell Biology, Tokyo Metropolitan Institute of Medical Science, Tokyo 156-8506, Japan; sanada-tk@igakuken.or.jp (T.S.); kohara-mc@igakuken.or.jp (M.K.); 3Toko Yakuhin Kogyo Co., Ltd., Osaka 530-0022, Japan; t.miyazaki@toko-yakuhin.co.jp (T.M.); masao_koshiba@toko-yakuhin.co.jp (T.K.); 4Vaccine Division, Biomedical Research Department, Center for Genetic Engineering and Biotechnology, Havana 10600, Cuba; julio.aguilar@cigb.edu.cu (J.C.A.R.); gerardo.guillen@cigb.edu.cu (G.G.N.)

**Keywords:** hepatitis B virus, cellular immunity, immunoglobulin A, infectious disease, carboxyl vinyl polymer

## Abstract

Hepatitis B vaccine induces the production of antibodies against hepatitis B surface antigen (anti-HBs) and prevents hepatitis B virus (HBV) infection. However, 5–10% of individuals cannot develop anti-HBs even after multiple vaccinations (HB vaccine non-responders). We developed an intranasal vaccine containing both HBs antigen (HBsAg) and HB core antigen (HBcAg) and mixed it with a viscosity enhancer, carboxyl vinyl polymer (CVP-NASVAC). Here, we investigated the prophylactic capacity of CVP-NASVAC in HB vaccine non-responders. Thirty-four HB vaccine non-responders were administered three doses of intranasal CVP-NASVAC. The prophylactic capacity of CVP-NASVAC was assessed by evaluating the induction of anti-HBs and anti-HBc (IgA and IgG) production, HBV-neutralization activity of sera, and induction of HBs- and HBc-specific cytotoxic T lymphocytes (CTLs). After CVP-NASVAC administration, anti-HBs and anti-HBc production were induced in 31/34 and 27/34 patients, respectively. IgA anti-HBs and anti-HBc titers significantly increased after CVP-NASVAC vaccination. HBV-neutralizing activity in vitro was confirmed in the sera of 26/29 CVP-NASVAC-administered participants. HBs- and HBc-specific CTL counts substantially increased after the CVP-NASVAC administration. Mild adverse events were observed in 9/34 participants; no serious adverse events were reported. Thus, CVP-NASVAC could be a beneficial vaccine for HB vaccine non-responders.

## 1. Introduction

The World Health Organization reports more than 250 million individuals are chronically infected with hepatitis B (HB) virus (HBV) globally [1]. Chronic HBV infection causes chronic hepatitis, liver cirrhosis, liver failure, and hepatocellular carcinoma. In addition, 0.8 million people die of HBV-related diseases annually. HBV infection is regarded as one of the most serious infectious diseases worldwide. Interferon and nucleos(t)ide analogs are the two main HBV treatment drugs; however, HBV cannot be completely eliminated using the currently available drugs. Therefore, prevention of HBV infection using the HB vaccine is the best strategy to reduce the incidence of HBV infection and related mortality, as it induces the production of antibodies against HB surface antigens (anti-HBs) [2]. However, approximately 5–10% of immunized individuals cannot produce anti-HBs even after multiple doses of HB vaccination; they are called HB vaccine non-responders [3]. The following are considered risk factors for non-response to HB vaccines: genetic predisposition, chronic liver diseases, chronic kidney diseases, immunomodulatory medications, human immunodeficiency virus infection, older age, male sex, obesity, smoking, and vaccine administration in the buttock [4,5,6]. Although HB vaccine non-responders are at risk of HBV infection, no fundamental solution is available.

The conventional HB vaccine contains only HB surface antigen (HBsAg) as an immunogen for inducing anti-HBs production [7]. We developed a vaccine containing both HBsAg and HB core antigen (HBcAg) (NASVAC) that can be nasally administered [8]. NASVAC was designed to induce anti-HBs production for HBV neutralization and HBcAg-specific cytotoxic T lymphocyte (CTL) production to inhibit HBV replication. HBcAg acts as an adjuvant for HBsAg and amplifies HBsAg-specific immune responses [8]. The nose is an attractive site for immunization because nasal vaccination induces stronger immune responses than subcutaneous or intramuscular vaccination [9]. In addition, intranasal immunization introduces IgA-type antibodies to all mucosae, including the nasal, lung, gut, and sexual gland mucosae [10]. Previously, we conducted a phase III clinical trial of NASVAC in treatment-naive patients with chronic HBV infection in Bangladesh. NASVAC displayed a superior reduction in HBV DNA level compared to peg-IFN. To increase the immunogenicity of NASVAC, it was mixed with carboxyl vinyl polymer (CVP), filled into a dedicated device for nasal administration, and named CVP-NASVAC. CVP increases the viscosity of the vaccine solution, enables adherence to the nasal mucosa, and prolongs retention time in the nasal cavity [11]. Furthermore, the dedicated device sprays the vaccine at a uniform angle and density while maintaining viscosity. CVP-NASVAC elicited stronger anti-HBV immune responses than the conventional HB vaccine [12].

In our previous clinical trial, we successfully induced anti-HBs production in approximately 50% of the patients with chronic HBV infection by administering CVP-NASVAC [13]. Therefore, we hypothesized that CVP-NASVAC induces anti-HBs production in HB vaccine non-responders. In this study, we aimed to investigate the prophylactic capacity of CVP-NASVAC in HB vaccine non-responders.

## 2. Materials and Methods

### 2.1. Study Population and Protocol

This study was conducted in the Department of Gastroenterology and Metabology, Ehime University Graduate School of Medicine, Toon, Japan. The study protocol complied with the 2013 Declaration of Helsinki, was approved by the institutional review board of Ehime University (E18-28) and was registered in a clinical trials registry in Japan (#jRCTs061180101). HB vaccine non-responder was defined as an individual with a history of conventional HB vaccination, anti-HBs serum titer <10 mIU/mL, and anti-HBc <1.0 cut-off index (C.O.I.). Participants were enrolled after they provided written informed consent. The NASVAC formulation was a mixture of *Pichia pastoris*-derived recombinant HBsAg subtype adw2 and *Escherichia coli*-derived recombinant full-length HBcAg. NASVAC was produced at the Center for Genetic Engineering and Biotechnology (CIGB, Havana City, Cuba). An 800-μL of NASVAC (containing HBsAg: 80 μg, HBcAg: 80 μg) was mixed with 200 μL of CVP (Toko Yakuhin Kogyo Co., Ltd., Osaka, Japan). The participants were administered intranasal CVP-NASVAC fortnightly for a total of three times. Blood samples were collected before 1, 3, and 6 months after the final administration of CVP-NASVAC.

### 2.2. IgG-Type Anti-HBs and IgG-Type Anti-HBc and Anti-HBe Measurements

Serum levels of IgG-type anti-HBs (Fujirebio Inc., Tokyo, Japan) and IgG-type anti-HBc (Fujirebio Inc.) were measured using Lumipulse L2400 (Fujirebio Inc.) with chemiluminescent enzyme immunoassay (CLEIA) according to the manufacturer’s protocol. Anti-HBs in the serum samples were incubated with alkaline phosphatase-conjugated HBsAg (adr, genotype C) and HBsAg-binding ferrite particles. After washing with phosphate-buffered saline (PBS), 3-(2′-spiroadamantane)-4-methoxy-4-(3″-phosphoryloxy) phenyl-1, 2-dioxetane disodium salt was added to the composites to measure chemiluminescence and anti-HBs concentration was measured. Anti-HBc in serum samples was incubated with HBcAg-binding ferrite particles. After washing with PBS, the composites were incubated with alkaline phosphatase-conjugated anti-human IgG; then, 3-(2′-spiroadamantane)-4-methoxy-4-(3″-phosphoryloxy) phenyl-1, 2-dioxetane disodium salt was added to measure chemiluminescence. The C.O.I of anti-HBc was calculated as the ratio of the amount of luminescence emitted in the sample to that emitted in the positive control. Serum levels of anti-HBe (Abbott Laboratories, Chicago, IL, USA) were measured using ARCHITECT i2000SR (Abbott Laboratories) with chemiluminescent immunoassay (CLIA) according to the manufacturer’s protocol. The cut-off value for anti-HBe negativity of inhibition rate is less than 60% compared to the control.

### 2.3. IgA-Type Anti-HBs and Anti-HBc Detection Using Enzyme-Linked Immunosorbent Assay 

U96-Nunc-Maxisorp plates (Thermo Fisher Scientific, Waltham, MA, USA) were coated with 50 µL of HBs-S (genotype C) or HBc protein (Beacle, Inc., Kyoto, Japan) diluted to 2 µg/mL in 50 mM carbonate buffer (pH 9.6) per well. 200 µL of blocking buffer (PBS containing 1% (*w*/*v*) bovine serum albumin, 0.5% (*v*/*v*) Tween 20, and 2.5 mM ethylenediaminetetraacetic acid) was added after the 18 h incubation at 4 °C. Then plates were incubated for 2 h at 22 °C. After incubation, the plates were washed five times with 200 µL of PBS containing 0.05% (*v*/*v*) Tween 20 (PBST). Next, each serum sample diluted 1:100 in blocking buffer was dispensed into the plates at 50 µL per well. After incubation at 22 °C for 2 h, the plates were washed five times with PBST. Next, biotin-conjugated anti-human IgA (Abcam, Cambridge, UK) diluted to 1 µg/mL or anti-human IgG (SouthernBiotech, Birmingham, AL, USA) diluted to 2 ng/mL in blocking buffer was added 50 µL per well, and the plate was incubated for 1 h at 22 °C. After the five times washing with PBST, streptavidin horseradish peroxidase (Vector Laboratories Inc., Burlingame, CA, USA) diluted to 1 µg/mL in blocking buffer, 50 µL was added per well. After 1 h incubation at 22 °C, the plates were washed with PBST five times. Next, 100 µL of o-phenylenediamine substrate diluted to 400 µg/mL in hydrogen peroxide was added to the wells, and the plates were incubated for 10 min at 22 °C. The reaction was ended by adding 2 M sulfuric acid for each well, and the absorbance was measured at 492 nm (Asys Expert 96; Harvard Bioscience Inc., MA, USA).

### 2.4. HBV-Neutralization Test

HepG2-hNTCP-30 cells were obtained from the Tokyo Metropolitan Institute of Medical Science (Tokyo, Japan). The cells were cultured in Dulbecco’s modified Eagle medium (DMEM; Nissui, Tokyo, Japan) supplemented with 10% (*v*/*v*) fetal calf serum (FCS), 10 µg/mL L-glutamine, 10 µg/mL non-essential amino acids, 0.2% (*v*/*v*) NaHCO_3_, 100 U/mL penicillin G, and 100 µg/mL streptomycin. Next, 5.0 × 10^5^ HepG2-hNTCP-30 cells per well were plated in collagen-coated 48-well plates and maintained in DMEM/F12 + GlutaMAX (Thermo Fisher Scientific) supplemented with 10% (*v*/*v*) FCS, 10 mM 4-(2-hydroxyethyl)-1-piperazineethanesulfonic acid, 5 µg/mL insulin, 100 U/mL penicillin G, 100 µg/mL streptomycin, 0.4 µg/mL puromycin, and 2% (*v*/*v*) dimethyl sulfoxide. Serum samples from non-responders were heat-inactivated for 30 min at 56 °C and then subjected to two-fold serial dilutions in culture medium in 96 well plates. The dilutions were combined with an equal volume of an HBV inoculum containing 3.75 × 10^5^ viral DNA copies (PhoenixBio Co., Ltd., Higashi-Hiroshima, Japan). The mixture of serum and HBV was incubated for 1 h at 37 °C. Thereafter, 125 µL of each mixture was inoculated to HepG2-hNTCP-30 cell (7.5 GEq/cell). After the 48 h incubation at 37 °C, the cells were washed five times with the culture medium and collected. HBV DNA was extracted from the cell pellets using SMITEST EX-R & D (MBL, Tokyo, Japan) and then used for qPCR analysis. The neutralizing antibody titer of each serum sample was determined as the reciprocal of the maximum dilution of serum that reduced the viral DNA level by 90% or more compared to the control sample.

### 2.5. HBV-Specific CTL Detection Using Enzyme-Linked Immunosorbent Spot Assay

HBV-specific CTLs were detected using commercial enzyme-linked immunosorbent spot (ELISPOT) assay kits (Human IFN-γ ELISpot PLUS; MABTECH, Stockholm, Sweden). Briefly, 2 × 10^5^ peripheral blood mononuclear cells (PBMCs) from non-responders were harvested in 96-well plates containing Roswell Park Memorial Institute 1640 medium supplemented with 10% (*v*/*v*) FCS, 100 U/mL penicillin G, and 100 µg/mL streptomycin. The cells were further stimulated with either HBsAg (genotype C) or HBcAg (Beacle, Inc.) or anti-CD3 monoclonal antibodies as the positive control. After 72 h of incubation at 37 °C under 5% CO_2_, 0.1 µg/well of the detection antibody was added to the samples; then, the plates were incubated for another 2 h at 22 °C. Subsequently, streptavidin horseradish peroxidase was added to the samples and incubated at 22 °C for 1 h. Finally, 100 μL/well of 3,3′,5,5′-Tetramethylbenzidine substrate solution was added to the samples, and the plates were incubated for another 5–10 min at 22 °C. The plates were washed with PBS, and spots were analyzed using the AID ELISPOT Reader 08 classic (Advanced Imaging Devices GmbH, Straßberg, Germany).

### 2.6. Statistical Analysis

Statistical analyses were performed using JMP version 13.0.0 (SAS Institute Inc., Cary, NC, USA). The Wilcoxon signed-rank test was used to analyze the data. Bivariate correlations were determined using Pearson’s correlation coefficient and the test of no correlation. Statistical significance was defined by *p* < 0.05.

## 3. Results

### 3.1. Study Profile and Participant Demographics

Seventy-eight healthy individuals were enrolled in this study after they provided written consent; of them, 40 were identified as HB vaccine non-responders. Six individuals were excluded because of missing data. Eventually, 34 HB vaccine non-responders were included in the analysis (Figure 1). The demographic characteristics of the participants are presented in Table 1. The median age was 23 years, and the number of male and female individuals was 31 and 3, respectively. All the participants received at least one series of standard vaccinations, 2 received two series of vaccinations, and three received three series of vaccinations. The median duration from the last HB vaccination to enrollment was 2 years. The anti-HBs titer was 0 mIU/mL in 22 non-responders and 0–10 mIU/mL in 12 non-responders. Anti-HBc and anti-HBe antibodies were absent in all participants. All participants received three doses of CVP-NASVAC (one dose every 2 weeks). Adverse events were observed in 9/34 (26.5%) non-responders, but no serious events were reported during or after CVP-NASVAC administration. The most frequent adverse event was an increase in the C-reactive protein level (9/34, 26.5%). The remaining events are listed in Table 2.

### 3.2. Induction of IgG-Type Anti-HBs and Anti-HBc Production by CVP-NASVAC in HB Vaccine Non-Responders

Serum samples from non-responders were analyzed to investigate IgG antibody responses using CLEIA. Before CVP-NASVAC administration, anti-HBs positivity was 0% (*n* = 0/34), and the mean titer was 1.86 ± 2.84 mIU/mL. One month after three doses of CVP-NASVAC vaccination, 31/34 (91.1%) individuals were positive for IgG-type anti-HBs with an average titer of 2124.34 ± 2850.1 mIU/mL (Figure 2a). Anti-HBs positivity was maintained in 30/34 (88.2%) non-responders 6 months after CVP-NASVAC vaccination, and the mean titer was 926.79 ± 1730.33 mIU/mL (Figure 2a).

None of the HB vaccine non-responders were positive for IgG-type anti-HBc before CVP-NASVAC administration. IgG-type anti-HBc was positive in 27/34 individuals (79.4%, mean titer: 10.58 ± 10.89 C.O.I.) 1 month after CVP-NASVAC administration, and the positivity was maintained in 24/34 individuals (70.6%, mean titer: 6.18 ± 6.65 C.O.I.) after 6 months (Figure 2b).

### 3.3. Induction of IgA-Type Antibody Production by CVP-NASVAC

To investigate the effect of intranasal vaccination on mucosal immunity, we evaluated the IgA-type antibody response using an enzyme-linked immunosorbent assay. Sera from five participants were not stored; thus, the sera from the remaining 29 participants were analyzed. The serum levels of IgA anti-HBs and anti-HBc before vaccination and 1 month after vaccination were compared; 21/29 (72.4%) HB vaccine non-responders showed an increase in IgA-type anti-HBs production after CVP-NASVAC administration (Figure 3a). Similarly, an elevation in IgA-type anti-HBc level was observed in 28/29 (96.5%) HB vaccine non-responders after CVP-NASVAC administration (Figure 3b).

### 3.4. HBV Neutralization by the Antibody in the Sera of CVP-NASVAC-Immunized HB Vaccine Non-Responders

To evaluate the HBV-neutralizing activity of the antibody induced by CVP-NASVAC, we conducted an in vitro HBV-neutralization test. The sera from five participants were not stored properly; thus, the sera from the remaining 29 participants were analyzed. Although none of the HB vaccine non-responders showed HBV neutralization before CVP-NASVAC administration, 26/29 (89.6%) participants showed HBV-neutralizing activity after CVP-NASVAC administration (Figure 4a). The neutralizing activity was significantly correlated with the titer of IgG-type anti-HBs (Figure 4b). These results indicate that CVP-NASVAC induced anti-HBs production in HB vaccine non-responders, neutralizing HBV particles.

### 3.5. Induction of HBcAg-Specific CTL Production by CVP-NASVAC

HBcAg-specific CTLs, which inhibit HBV replication, were detected as HBcAg-specific IFN-γ-producing cells using an ELISPOT assay. When performing the ELISPOT assay, 7/34 (20.6%) PBMC samples of non-responders were excluded from the analysis because of poor cell conditions. The results are shown in Figure 5, and HBsAg- and HBcAg-specific IFN-γ-producing cells are shown as spots (Figure 5a). In total, 25/27 (92.6%) HB vaccine non-responders showed a significant increase in the HBcAg-specific CTL count 6 months after CVP-NASVAC administration compared with that before vaccination (Figure 5b). Similarly, 25/27 (92.6%) HB vaccine non-responders showed an increase in the HBsAg-specific CTL count after CVP-NASVAC administration compared with that before vaccination (Figure 5c). These findings demonstrate that CVP-NASVAC-induced HBsAg- and HBcAg-specific cellular immune responses were similar to HBsAg- and HBcAg-specific humoral immune responses.

## 4. Discussion

The estimated prevalence of HB vaccine non-responders, a population at risk of HBV infection, is approximately 5–10%; however, no fundamental solution for HB vaccine non-responders is currently available. In this study, we investigated the prophylactic capacity of CVP-NASVAC in HB vaccine non-responders and demonstrated that CVP-NASVAC induced HBsAg/HBcAg-specific humoral and cellular immune responses. Over 90% of HBV vaccine non-responders acquired IgG-type anti-HBs after CVP-NASVAC vaccination. Furthermore, anti-HBs-acquired serum after CVP-NASVAC administration displayed HBV neutralization in vitro. Thus, CVP-NASVAC could serve as a prophylactic vaccine for HB vaccine non-responders at risk of HBV infection.

HBsAg and HBcAg are viral-like particles of size 20–30 nm. HBcAg acts as an adjuvant and enhances HBsAg-specific immune responses. In animal experiments, we found that anti-HBs titers were higher when HBsAg was co-injected with HBcAg than when only HBsAg was injected [8]. The nose is an ideal site for immunization [14]. The serum titer of induced IgG-type anti-HBs after nasal immunization is higher than that after subcutaneous or intramuscular immunization [12]. Intranasal immunization induces the production of both IgG- and IgA-type antibodies, whereas subcutaneous administration induces the production of only IgG-type antibodies [10,12]. Antigens administered via the nose are taken up by antigen-presenting cells (APCs) in mucosa-associated lymphoid tissue, and antigen-specific T and B cells are activated to produce secretory IgA and IgM. Immune cells activated in one mucosal tissue induce systemic mucosal immunity, and IgA antibodies also exhibit neutralizing activity [15]. In this study, we found that IgA-type anti-HBs and anti-HBc titers significantly increased in non-responders after CVP-NASVAC administration.

Nasal immunization induces anti-HBs production in the mucosa, nose, lung, intestine, and sex glands [10]. HBV is transmitted through sexual contact; therefore, IgA-type anti-HBs in the sex glands might effectively prevent HBV infection. CVP increases the viscosity of the vaccine and enables the vaccine to adhere to the mucosal surface of the upper respiratory tract, where many APCs exist. Thus, CVP prolongs the retention time of vaccines in the nasal cavity. In an experiment in rhesus monkeys, a vaccine mixed with CVP remained in the nasal cavity for over 6 h [11]. A special device for nasal administration was designed to spray a highly viscous vaccine plus CVP of particle size 50–100 µm, which is suitable for mucosal attachment and APC uptake. Furthermore, the device sprays the vaccine across a wide area, from the inferior to the superior turbinate.

HBV neutralization was confirmed using an in vitro neutralizing assay. Here, none of the participants displayed HBV neutralization before the CVP-NASVAC administration; however, nearly 90% of HB vaccine non-responders gained HBV-neutralization capacity after the CVP-NASVAC administration. Remarkably, the HBV-neutralizing activity positively correlated with the titer of IgG-type anti-HBs. In contrast, in vitro neutralization of HBV could not be achieved in any of the three participants who failed to produce anti-HBs after CVP-NASVAC administration. On the other hand, anti-HBs titer decreased to below 100 mIU/mL in some participants during 6 months of follow-up. Previous papers suggested that an anti-HBs titer of 100 mIU/mL is the actual protective concentration [16,17]. We should follow anti-HBs titer and check the HBV neutralizing capacity.

HBcAg-specific CTLs play an important role in HBV elimination by inhibiting HBV replication [18,19]. In this study, we demonstrated that CVP-NASVAC induces HBcAg-specific CTL production. The number of HBcAg-specific CTLs significantly increased in over 90% of HB vaccine non-responders after CVP-NASVAC administration. HBcAg-specific CTL production induced by CVP-NASVAC may contribute to HBV prophylaxis similar to anti-HBs.

Our study has several limitations. First, we could not verify the prophylactic capacity of CVP-NASVAC in an in vivo HBV infection experiment. No suitable small animal model with a normal immune system is currently available for studying HBV infection [20]. Therefore, we investigated anti-HBs induction in vivo, neutralizing activity in vitro, and HBcAg-specific CTL detection ex vivo instead of conducting in vivo HBV infection experiments. Second, we could not investigate anti-HBs in the mucosal tissues, including the nasal and vaginal mucosa. That is, we investigated the anti-IgA response using plasma samples but not mucosal samples. A previous study on nasal vaccines reported that serum IgA antibody titers correlate with nasal IgA titers and that nasal neutralizing titers correlate with anti-IgA titers [21]. Third, we conducted an HBV-neutralizing assay using the HBV genotype C. In the future, we need to test the neutralizing activity of other genotypes. Finally, a selection bias may have influenced our results as the study population included hospital staff and medical students. These participants are not representative of the entire population.

Nevertheless, our study also has strengths. There are several reasons for the successful induction of anti-HBs production by CVP-NASVAC in HB vaccine non-responders. First, CVP-NASVAC contains two antigens: HBsAg and HBcAg. Second, CVP-NASVAC was administered intranasally and activated mucosal immunity. Third, CVP increased the retention time of NASVAC in the nasal mucosa. Finally, a special device was designed for nasal administration.

In conclusion, we report the safety of CVP-NASVAC and demonstrate that CVP-NASVAC induces anti-HBs, anti-HBc, IgA-type anti-HBs, and HBV-specific CTL production in HB vaccine non-responders. Thus, CVP-NASVAC can be an effective strategy for HBV prophylaxis in HB vaccine non-responders.

## Figures and Tables

**Figure 1 vaccines-11-01479-f001:**
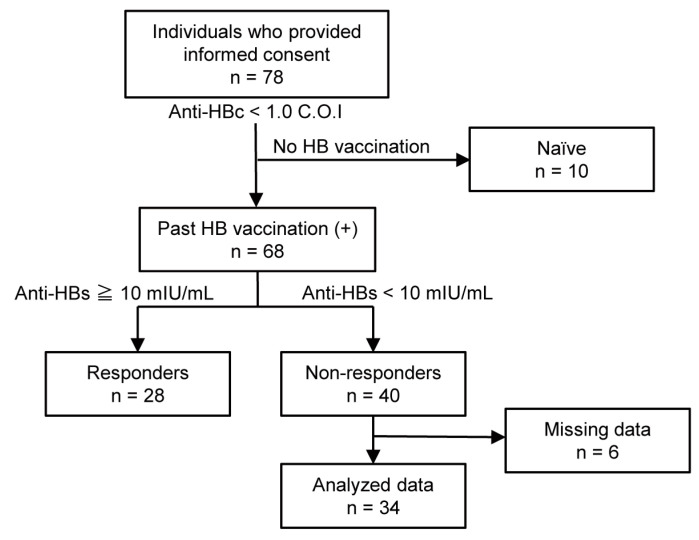
Clinical trial profile. Anti-HBs, the antibody against hepatitis B surface antigen; anti-HBc, the antibody against hepatitis B core antigen; C. O. I., cut-off value.

**Figure 2 vaccines-11-01479-f002:**
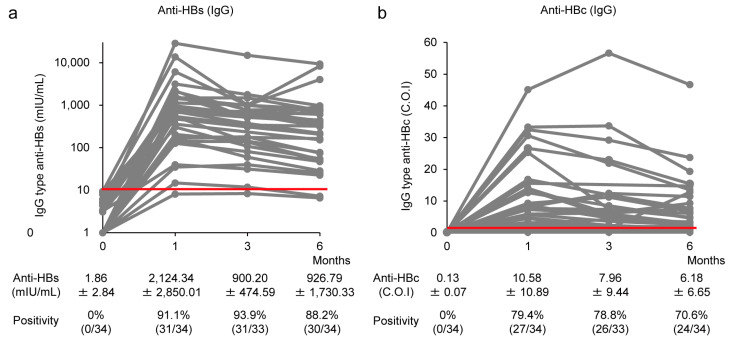
Induction of IgG-type anti-HBs and anti-HBc production by CVP-NASVAC. The titers of IgG-type (**a**) anti-HBs and (**b**) anti-HBc for each HB vaccine non-responder are plotted on the *y*-axis. Each gray line represents the titer of an individual. The red line represents the cut-off value for anti-HBs (10 mIU/mL) and anti-HBc positivity (1.0 C.O.I.). The titers are shown as mean ± standard deviation. CVP-NASVAC, an intranasal vaccine containing both hepatitis B surface and core antigens mixed with carboxyl vinyl polymer; anti-HBs, the antibody against hepatitis B surface antigen; anti-HBc, the antibody against hepatitis B core antigen; C.O.I., cut-off value.

**Figure 3 vaccines-11-01479-f003:**
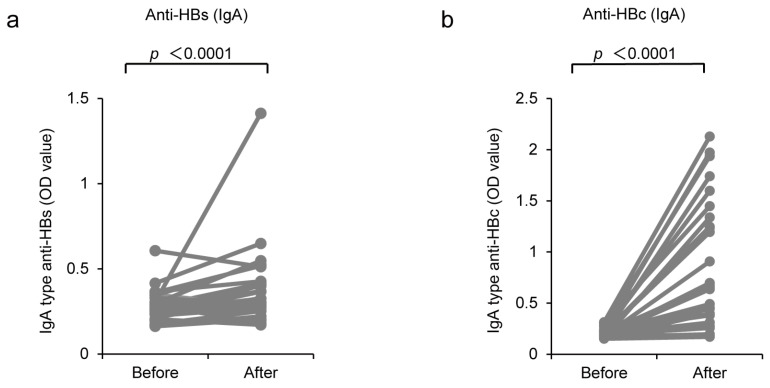
Induction of IgA-type anti-HBs and anti-HBc production by CVP-NASVAC. The OD values of IgG-type (**a**) anti-HBs and (**b**) anti-HBc for each HB vaccine non-responder are plotted on the *y*-axis. Each gray line represents the titer value of an individual. The p-values were calculated using the Wilcoxon signed-rank test. *p* < 0.05, compared before and after vaccination. CVP-NASVAC is an intranasal vaccine containing both hepatitis B surface and core antigens mixed with carboxyl vinyl polymer; anti-HBs, the antibody against hepatitis B surface antigen; anti-HBc, the antibody against hepatitis B core antigen.

**Figure 4 vaccines-11-01479-f004:**
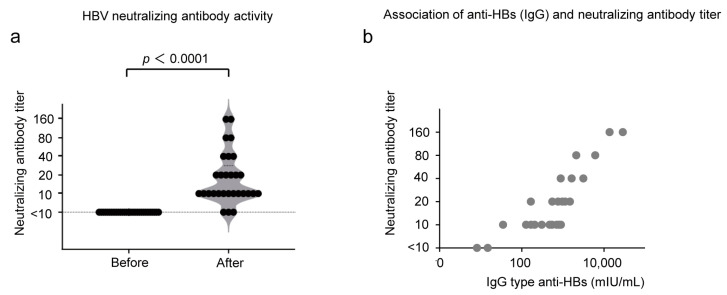
HBV neutralization by CVP-NASVAC-induced anti-HBs. (**a**) The neutralizing titer is plotted on the *y*-axis. Each black dot represents the titer value for each individual. The p-values were calculated using the Wilcoxon signed-rank test; *p* < 0.05, compared before and after vaccination. (**b**) The neutralizing antibody titer (*y*-axis) and IgG-type anti-HBs titer (*x*-axis) are plotted. Each gray dot represents each individual. Bivariate correlations were determined using Pearson’s correlation coefficient and the test of no correlation. CVP-NASVAC, an intranasal vaccine containing both hepatitis B surface and core antigens mixed with carboxyl vinyl polymer; anti-HBs, antibody against hepatitis B surface antigen; anti-HBc, antibody against hepatitis B core antigen; HBV, hepatitis B virus.

**Figure 5 vaccines-11-01479-f005:**
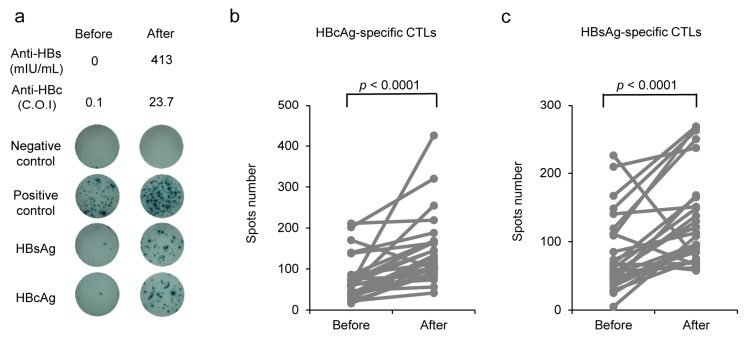
HBsAg- and HBcAg-specific CTL induction by CVP-NASVAC. (**a**) Representative image of enzyme-linked immunosorbent spot assay of a participant (HC098). The number of HBcAg- (**b**) and HBsAg- (**c**) specific IFN-γ-producing CTLs are plotted on the *y*-axis. The p-values were calculated using the Wilcoxon signed-rank test; *p* < 0.05 when compared before and after vaccination. CVP-NASVAC, an intranasal vaccine containing both hepatitis B surface and core antigens mixed with carboxyl vinyl polymer; anti-HBs, antibody against hepatitis B surface antigen; anti-HBc, antibody against hepatitis B core antigen; HBV, hepatitis B virus; CTL, cytotoxic T lymphocyte; HBsAg, hepatitis B surface antigen; HBcAg, hepatitis B core antigen.

**Table 1 vaccines-11-01479-t001:** Participant demographics.

Number of participants	34
Age (years)	23 (22–25)
Male: Female	31: 3
AST (U/L)	22 (19–26)
ALT (U/L)	19.5 (15–25)
Platelet count (×10⁴/µL)	23.8 (21.5–27.9)
Anti-HBs (mIU/mL)	0 (0–3.3)
Anti-HBc (C.O.I.)	0.1 (0.1–0.1)
Anti-HBe positivity (%)	0 (0%)
Duration from the last vaccination (years)	2 (2–3)

Data are presented as numbers and median (IQR). Abbreviations: IQR, interquartile range; AST, aspartate aminotransferase; ALT, alanine aminotransferase; HB, hepatitis B; C.O.I., cut-off index; anti-HBs, the antibody against hepatitis B surface antigen; anti-HBc, the antibody against hepatitis B core antigen; anti-HBe, antibody against hepatitis B e antigen.

**Table 2 vaccines-11-01479-t002:** Adverse events were observed during and after NASVAC treatment.

	n (%)
All AEs	9 (26.5%)
Increase in CRP level	3 (8.8%)
Nasal mucus formation	2 (5.9)
Sneezing	2 (5.9)
Fever	2 (5.9)
General fatigue	2 (5.9)
Stuffy nose	1 (2.9)
Pruritus	1 (2.9)
Increase in WBC count	1 (2.9)
Increase in ALT level	1 (2.9)

Abbreviations: AE, adverse event; CRP, C-reactive protein; WBC, white blood cell; ALT, alanine aminotransferase; NASVAC, vaccine containing both hepatitis B surface and core antigens.

## Data Availability

All data can be available from Osamu Yoshida, Ehime University Graduate School of Medicine, Ehime, Japan (yoshidao@m.ehime-u.ac.jp). He is the corresponding author of this paper.

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
