# Peer review of "Intranasal HBsAg/HBcAg-Containing Vaccine Induces Neutralizing Anti-HBs Production in Hepatitis B Vaccine Non-Responders"

_vaccines, 2023, doi:10.3390/vaccines11091479_

Round 1

Reviewer 1 Report

The study by Shiraishi et al., entitled “Intranasal HBsAg/HBcAg-containing vaccine induces neutralizing anti-HBs production in hepatitis B vaccine non-responders” investigated the prophylactic capacity of CVP-NASVAC in HB vaccine non-responders. A total of 34 HBV vaccine nonresponder were included in the study and participants were administered three doses of vaccine by intranasal route. The authors concluded that CVP-371 NASVAC induces anti-HBs, anti-HBc, IgA-type anti-HBs, and HBV-specific CTL production in HB vaccine non-responders. Thus, CVP-NASVAC can be an effective strategy for HBV prophylaxis in HB vaccine non-responders. This is well designed, interesting study, having the relevance for HBV vaccine nonresponder population.

Minor comments to Authors

1) In the material and methods section, authors have mentioned the definition of vaccine non responder (line 87 and 88), individual with a history of conventional HB vaccination. Authors need to mention here, how many time/doses of vaccine were given to the participants.

2) In the material and methods section, authors have mentioned, the participants were administered intranasal CVP-NASVAC fortnightly for a total of three times. Blood samples were collected before and 1, 3, and 6 months after CVP-NASVAC administration. What was the rationale for deciding the three doses and interval of dose?

3) In the table1, participant demographics, please explain what is the meaning of Anti-HBe(%inh)? If the authors have used The ARCHITECT Anti-HBe assay, then they should mention it in the material and method section with reference and cut off value for positivity or negativity.

4) In the figure 2a, page 7, for some of the participants anti-HBs level were below 100 throughout the course of analysis (from month one to month six), however appropriate immune response is defined as an HBsAb level of greater than 10 IU/L but, plenty of data suggest that an HBsAb level between 10 IU/L and 100 IU/L may indicate incomplete response to the HBV vaccine and places patients at risk of loss of immunity against HBV. An HBsAb level of at least 100 IU/L is considered protective. This needs clear explanation in the text.

5) Some of the sentences need proper English grammar correction and rephrasing.

Some of the sentences need proper English grammar correction and rephrasing.

Author Response

Response to Reviewers

Thank you very much for your valuable comments about our manuscript. Point to point responses of reviewers’ comment is provided in bolded type. Also, manuscript has been revised on the basis of your comments.

Reviewer #1

The study by Shiraishi et al., entitled “Intranasal HBsAg/HBcAg-containing vaccine induces neutralizing anti-HBs production in hepatitis B vaccine non-responders” investigated the prophylactic capacity of CVP-NASVAC in HB vaccine non-responders. A total of 34 HBV vaccine nonresponder were included in the study and participants were administered three doses of vaccine by intranasal route. The authors concluded that CVP-371 NASVAC induces anti-HBs, anti-HBc, IgA-type anti-HBs, and HBV-specific CTL production in HB vaccine non-responders. Thus, CVP-NASVAC can be an effective strategy for HBV prophylaxis in HB vaccine non-responders. This is well designed, interesting study, having the relevance for HBV vaccine nonresponder population.

Minor comments to Authors;

1. In the material and methods section, authors have mentioned the definition of vaccine non responder (line 87 and 88), individual with a history of conventional HB vaccination. Authors need to mention here, how many time/doses of vaccine were given to the participants.

In this study, all participants had received at least one series of vaccine schedule (three doses administrations), and 2/34 had received two series and 3/34 had three series. We have mentioned details of previous vaccination (line 190-192).

 2. In the material and methods section, authors have mentioned, the participants were administered intranasal CVP-NASVAC fortnightly for a total of three times. Blood samples were collected before and 1, 3, and 6 months after CVP-NASVAC administration. What was the rationale for deciding the three doses and interval of dose?

In this study, the participants received three doses of CVP-NASVAC every two weeks, then we collected blood samples before and 1, 3, 6 months after the final CVP-NASVAC administration. We decided our protocol according to our previous animal experiment13) and clinical trial of CVP-NASVAC14). In animal experiment, Tupaia developed anti-HBs after three doses of CVP-NASVAC administration13). In the clinical study, chronic hepatitis B patients received CVP-NASVAC every two weeks, total ten doses, and the time of developing anti-HBs was around the third dose of administration. Therefore, we decided the protocol of CVP-NASVAC for prophylactic administration at three doses every two weeks. We corrected the sentence in the Material and Method section (line 96-97) .

3. In the table1, participant demographics, please explain what is the meaning of Anti-HBe(%inh)? If the authors have used The ARCHITECT Anti-HBe assay, then they should mention it in the material and method section with reference and cut off value for positivity or negativity.

Anti-HBe (%inh) is confusing. We changed “Anti-HBe(%inh)” to ”Anti-HBe positivity (%)” in table 1. We measured serum anti-HBe using ARCHITECT i2000SR (Abbott Laboratories)  according to manufacturer’s protocol. The cut off value for negativity of inhibition rate (%inh) is less than 60% compared to the control. We have added the sentence of anti-HBe protocol in method section (line 112-116).

4. In the figure 2a, page 7, for some of the participants anti-HBs level were below 100 throughout the course of analysis (from month one to month six), however appropriate immune response is defined as an HBsAb level of greater than 10 IU/L but, plenty of data suggest that an HBsAb level between 10 IU/L and 100 IU/L may indicate incomplete response to the HBV vaccine and places patients at risk of loss of immunity against HBV. An HBsAb level of at least 100 IU/L is considered protective. This needs clear explanation in the text.

As reviewer 1 mentioned, previous paper demonstrated that complete protectable anti-HBs level could be over 100 IU/L. Anti-HBs titer in some participants decreased to less than 100 mIU/mL during the follow-up period. Actually our data indicates that anti-HBs level below 100 mIU/mL displayed non or weaker neutralizing titer in vitro (Figure 4b). We should monitor anti-HBs titer in longer period and check the HBV neutralizing capacity. We mentioned in the discussion with 2 papers citation (line 351-354) .

5. Some of the sentences need proper English grammar correction and rephrasing.

We will correct the English grammar again using English proofreading.

Reviewer 2 Report

In this manuscript, the authors described the experiment of the administration of a novel HB vaccine that comprises both HBs and HBc antigens and is combined with a viscosity enhancer, carboxyl vinyl polymer. They indicated that their novel vaccine successfully induced humoral and cellular immunity even in the non-responders of the currently-used HB vaccine. The described data are interesting, but insufficient description or several errors are found. To improve the manuscript, the following issues should be addressed.

Major points

1. The authors used HBsAg and HBcAg in their novel HB vaccine, but there is no information about the genotype of antigens used. The HBV genotypes used for the vaccine, as well as the HBV genotypes of the antigens used to detect the vaccine-induced antibodies and the genotypes of the HBV antigens used for the ELISPOT assay, should be indicated. If the genotypes differ, homology information or alignment of the corresponding regions should be indicated. They mentioned that they used the HBV genotype C strain to evaluate the neutralizing activity of the induced antibody, but it is not described in Materials and Methods. It should be indicated.

2. To compare the neutralizing activity of the induced antibodies, the authors use the end-point dilution factor that reduces the HBV infection to less than 90%. This method is imprecise and uninformative. A dose-response curve should be indicated by using the neutralizing activity at each dilution of the induced antibody, and EC50 values should be given. Besides, the infection titer by treatment of the pre-immune serum should also be provided. Immunostaining of infected cells is better to be performed to demonstrate inhibition of HBV infection by induced antibodies.

3. As mentioned in Discussion, they used only the HBV genotype C strain to evaluate the neutralizing effects of the induced antibody. It is better to evaluate by using multiple HBV genotypes such as genotype D strain.

Minor points

       Line 191; 9/34 is not 8.8%.

       Table 1;

The meaning of ‘n’ is unknown.

The range of anti-HBs is indicated as ‘0 – 3.3’, and the average is indicated as ‘0’. Is it correct?

The description of anti-HBe by ‘%inh’ is confusing. It is better to be indicated by C.O.I. Or the range of ‘%inh’ when the antibody is positive should be indicated.

Author Response

Response to Reviewers

Thank you very much for your valuable comments about our manuscript. Point to point responses of reviewers’ comment is provided in bolded type. Also, manuscript has been revised on the basis of your comments.

Reviewer #2

In this manuscript, the authors described the experiment of the administration of a novel HB vaccine that comprises both HBs and HBc antigens and is combined with a viscosity enhancer, carboxyl vinyl polymer. They indicated that their novel vaccine successfully induced humoral and cellular immunity even in the non-responders of the currently-used HB vaccine. The described data are interesting, but insufficient description or several errors are found. To improve the manuscript, the following issues should be addressed.

Major issues:

1. The authors used HBsAg and HBcAg in their novel HB vaccine, but there is no information about the genotype of antigens used. The HBV genotypes used for the vaccine, as well as the HBV genotypes of the antigens used to detect the vaccine-induced antibodies and the genotypes of the HBV antigens used for the ELISPOT assay, should be indicated. If the genotypes differ, homology information or alignment of the corresponding regions should be indicated. They mentioned that they used the HBV genotype C strain to evaluate the neutralizing activity of the induced antibody, but it is not described in Materials and Methods. It should be indicated.

In this study, we used CLEIA for detecting IgG-type anti-HBs, which is commonly provided in daily clinic by a commercial testing company. Serotype adr (genotype C) derived HBsAg are used at this CLEIA. Although genotype difference might affect the results in anti-HBs titer in our CLEIA, we could measure anti-HBs in HB vaccine responders (anti-HBs >10 mIU/mL) who previously received either genotype A or C derived conventional HB vaccines. Thus, we think that genotype difference can be disregarded in IgG-anti-HBs measurement.

We used genotype C derived HBsAg for measuring IgA-type anti-HBs for ELISA. Because we have established our ELISA assay based on our animal experiment13). Ideally, we should use genotype A derived HBsAg for ELISA, fortunately we could confirm the increase of IgA-type anti-HBs/anti-HBc using genotype C derived HBsAg.

Genotype A derived HBsAg were used for PBMC stimulation in ELISPOT.

We added genotype information in material and method (CLEIA, ELISA, ELISPOT) (line 104, 120 and 167).

2. To compare the neutralizing activity of the induced antibodies, the authors use the end-point dilution factor that reduces the HBV infection to less than 90%. This method is imprecise and uninformative. A dose-response curve should be indicated by using the neutralizing activity at each dilution of the induced antibody, and EC50 values should be given. Besides, the infection titer by treatment of the pre-immune serum should also be provided. Immunostaining of infected cells is better to be performed to demonstrate inhibition of HBV infection by induced antibodies.

We could not calculate EC50. However, we created a supplemental figure which demonstrates the average HBV inhibition before and after CVP-NASVAC immunization in each serum dilutions. HBV inhibition by serum was depended on the serum titration.

We provided the neutralizing antibody titer of pre immunization serum in Figure 4a. No participants displayed HBV neutralization before CVP-NASVAC immunization. 

We did not performed immunostaining in this study, because it is difficult to stain HBV antigen in HBV infected HepG2-hNTCP-30 cells due to the low HBV replication in this cell line. So HepG2-hNTCP-30 cells is not suitable for monitoring HBV replication after the infection, but suitable for evaluating HBV internalization such as our neutralizing assay. On the other hand, PhoenixBio (PXB cell) might be suitable for immunostaining because HBV can internalize and replicate well in this cell. However, PXB cell is not cell line and we have to purchase this expensive cell in each experiment. Thus, we could not conduct HBV neutralizing experiment using PXB cells partially due to the financial reason.   

3. As mentioned in Discussion, they used only the HBV genotype C strain to evaluate the neutralizing effects of the induced antibody. It is better to evaluate by using multiple HBV genotypes such as genotype D strain.

We evaluated neutralizing activity using HBV genotype C strain because HBV of genotype C was commercially available. However, we could not obtain any genotype of HBV other than C. Ideally, we should conduct HBV neutralizing assay using all the genotype of HBV including genotype D. However, Kato M, et al, demonstrated that polyclonal anti-HBs acquired by either genotype A or C derived HB vaccine successfully prevented HBV infection of non-vaccine genotype, J Gastroenterol (2017) 52:1051–1063. Although neutralizing assay using only genotype C derived HBV is our limitation, we could demonstrate anti-HBs induction and HBV neutralization of genotype C HBV after the vaccination of genotype A derived CVP-NASVAC.

Minor points:

1. Line 191; 9/34 is not 8.8%.

We corrected 8.8% to 26.5% (line 198).

2. Table 1;

The meaning of ‘n’ is unknown.

The range of anti-HBs is indicated as ‘0 – 3.3’, and the average is indicated as ‘0’. Is it correct?

The description of anti-HBe by ‘%inh’ is confusing. It is better to be indicated by C.O.I. Or the range of ‘%inh’ when the antibody is positive should be indicated.

“n” is our mis wiring. We have deleted “n” from table 1.

In the range of anti-HBs, “0” means median of anti-HBs.

Anti-HBe (%inh) is confusing. We changed “Anti-HBe(%inh)” to ”Anti-HBe positivity (%)” in table 1. We measured anti-HBe by CLIA using ARCHITECT i2000SR (Abbott Laboratories)  and the cut off value for negativity of inhibition rate is less than 60% compared with control. We changed term in Table 1 and added the sentence of anti-HBe protocol in method section (line 112-116).

Round 2

Reviewer 2 Report

The revised version of the manuscript was improved by addressing the reviewer's comments. Now, it seems to be acceptable to publish.